# Genomic Diversity and Selection Signatures for Zaosheng Cattle

**DOI:** 10.3390/biology14060623

**Published:** 2025-05-28

**Authors:** Jianfeng Xu, Yanyan Wang, Fuyue Shi, Hailong Guo, Bo Gao, Junxiang Yang, Lingrong Gu, Dezhi Yang, Fengtao Zhang, Dengwei Gao, Ziyue Gao, Shengming Wang, Jin Wang

**Affiliations:** 1Animal Husbandry and Veterinary Research Institute of Gansu Province, Pingliang 744000, China; xjf_fx@126.com (J.X.); wyy19892025@126.com (Y.W.); 18009338630@163.com (B.G.); yjx20252025@163.com (J.Y.); gulingrong2022@sina.com (L.G.); 18864946960@163.com (Z.G.); w1394443437@163.com (S.W.); 2Animal Husbandry and Agriculture Technology Promotion Center, Qingyang 745000, China; gs_yangdezhi@126.com; 3Animal Husbandry and Veterinary Station, Ningxian 745200, China; zhangfengtao-1@163.com; 4Animal Husbandry Technology Promotion Center, Huanxian 745700, China; 13830408825@163.com; 5Institute of Gansu Pingliang Red Cattle, Pingliang 744000, China; marswww@163.com

**Keywords:** whole-genome sequencing, genetic diversity, selection signatures, local ancestry

## Abstract

Zaosheng cattle, a native Chinese breed from Gansu Province, possess excellent meat quality and resistance to heat and humidity. Here, we used whole-genome sequencing data from 110 samples to investigate the population structure, genomic diversity, and potential positive selection signals in Zaosheng cattle. Our results are important for the genetic improvement and resource conservation of Zaosheng cattle.

## 1. Introduction

In ancient China, cattle were used for plowing fields and sacrifices and, thus, held a preeminent status among the diverse range of domesticated species. Cattle can be categorized into the following two subspecies: humpless taurine (*Bos taurus*) and humped indicine (*B. indicus*) [1]. With the application of whole-genome sequencing technology, it has become possible to study the genomic genetic diversity of domestic animals, which is crucial for the genetic improvement of livestock and the conservation of local breeds. Previously, some scholars proposed that domestic cattle worldwide can be divided into the following five core groups: European taurine, Eurasian taurine, East Asian taurine, East Asian indicine, and South Asian indicine cattle [2]. Native Chinese cattle include Eurasian taurine, East Asian taurine, and East Asian indicine cattle. Recently, whole-genome sequencing technology has been widely applied to explore the population structure, genetic diversity, and selection signatures of livestock, including pigs [3], horses [4], cattle [2,5], sheep [6], and chickens [7]. Whole-genome resequencing, ancestral fragment inference, selective sweep analysis, and transcriptomic analysis have successively revealed characteristics of Sanjiang cattle and several genes linked to lipid metabolism, immune regulation, and stress reactions across the mosaic genome of Sanjiang cattle; these analyses indicate an excess of taurine or indicine ancestral features [8].

Zaosheng cattle are found primarily in Qingyang, Gansu Province. As a region with a predominantly agrarian culture, the local people have a long history of raising cattle [9]. Currently, the population of purebred Zaosheng cattle is relatively small. However, owing to their excellent meat quality, local farmers commonly use Zaosheng cattle as parent stock for crossbreeding [9]. Historical records show that people in the Zaosheng area have been selectively breeding Qinchuan cattle from the Guanzhong area of Shaanxi Province since 490 AD [9]. This has led to the formation of a larger local population of cattle. Recent studies have demonstrated a close genetic relationship between Zaosheng cattle and Qinchuan cattle [9], and some studies have explored the ancestral origins of Zaosheng cattle. Furthermore, Pingliang red cattle display a more prominent *B. taurus* pedigree in comparison with Zaosheng cattle [10]. However, there has been no in-depth analysis of the selection signatures in Zaosheng cattle or the contributions of different ancestral lineages to its genome. This was the focus of our study, which helped us understand the evolutionary history and ancestry of Zaosheng cattle and will provide a reference for resource conservation and the selective breeding of Chinese native cattle.

Selection signatures refer to genomic regions exhibiting reduced genetic diversity or distinctive allele frequency patterns caused by natural or artificial selection, providing critical insights into adaptive evolution [11]. In the present study, we used whole-genome resequencing data for 110 cattle, including newly sequenced data from 19 Zaosheng cattle. The data, which includes sequences from both commercial and native cattle breeds worldwide, were intergrated to facilitate comprehensive comparative genomic analyses. The key innovation of our study is the identification of genomic regions that have undergone positive selection and may be linked to adaptive traits that are beneficial for the survival and excellent meat quality of Zaosheng cattle in their native environments. Furthermore, we precisely traced the ancestral origins of these genomic regions. By pinpointing these positively selected regions, we have identified candidate genomic regions that uncover key genomic features that contribute to unique adaptations of Zaosheng cattle. Furthermore, our analyses revealed distinct ancestral contributions: the East Asian indicine lineage was found to predominantly influence growth-related traits (mediated by *PROKR1*), while the East Asian taurine lineage primarily contributed to meat quality characteristics (associated with *TP53INP2*). These findings are expected to offer valuable insights into the genetic basis of adaptive traits in Zaosheng cattle.

## 2. Materials and Methods

### 2.1. Sample Preparation and DNA Sequencing

To study the genetic diversity of Zaosheng cattle in China, we collected 19 samples of Zaosheng cattle from Gansu Province, all unrelated and sourced from the same farm (Appendix A). We adopted a standard phenol–chloroform method to extract genomic DNA from methylated preserved ear tissue. For each individual, paired-end sequencing data were obtained with an average insert size of 500 bp and an average read length of 150 bp via the Illumina NovaSeq system. The sequencing platform was from Novogene Bioinformatics Institute, Beijing, China. To compare the differences, we also used the whole genome sequences of 20 Central Chinese cattle (20 Qinchuan), 20 European taurine cattle (10 Angus and 10 Hereford), 20 East Asian taurine cattle (6 Yanbian and 14 Hanwoo), 20 East Asian indicine cattle (4 Guangfeng, 4 Ji’an, 7 Wenshan, and 5 Wannan), and 11 South Asian indicine (Appendix A).

### 2.2. Read Mapping and Variant Calling

We generated genotype data following the 1000 Bull Genomes Project Run 8 guidelines (https://research.wur.nl/en/datasets/run8-the-1000-bull-genomes-project accessed on 1 December 2024) (Appendix A). We removed low-quality bases and artifact sequences using Trimmomatic v0.39 [12], and all the clean reads were mapped to the taurine reference assembly (ARS-UCD1.2) and Btau_5.0.1 Y using BWA-MEM v0.7.13-r1126 with default parameters [12]. We then used SAMtools v1.9 [13] to sort bam files. For the mapped reads, potential PCR duplicates were identified using “MarkDuplicates” in Picard v2.20.2 (http://broadinstitute.github.io/picard accessed on 1 December 2024) The “BaseRecalibrator” and “PrintReads” of the Genome Analysis Toolkit (GATK, v.3.8-1-0-gf15c1c3ef) were used to perform base quality score recalibration (BQSR) with the known variant file (ARS1.2PlusY_BQSR_v3.vcf.gz) provided by the 1000 Bull Genomes Project.

For single-nucleotide polymorphism (SNP) calling, we created GVCF files using “HaplotypeCaller” in GATK with the “-ERC GVCF” option. We called and selected candidate SNPs from these combined GVCF files using “GenotypeGVCFs” and “SelectVariants”, respectively. To avoid possible false-positive calls, we used VariantFiltration of GATK as recommended by GATK best practices, as follows: (1) SNP clusters with “-clusterSize 3” and “-clusterWindowSize 10” options; (2) SNPs with mean depth (for all samples) < 1/3× and >3× (×, overall mean sequencing depth across all SNPs); (3) quality by depth, QD  <  2; (4) Phred-scaled variant quality score, QUAL  <  30; (5) strand odds ratio, SOR  >  3; (6) Fisher strand, FS  >  60; (7) mapping quality, MQ  <  40; (8) mapping quality rank sum test, MQRankSum < −12.5; and (9) read position rank sum test, ReadPosRankSum < −8 were filtered. We then filtered out nonbiallelic SNPs and SNPs with missing genotype rates > 0.1. The imputation and phasing of the SNPs were simultaneously performed using BEAGLE v4.0 with default parameters, and the SNPs were filtered with DR2  <  0.9. The remaining SNPs were annotated according to their positions using ANNOVAR 2020Jun08 version (http://www.openbioinformatics.org/annovar/, accessed on 1 December 2024) [14].

### 2.3. Population Genetic Analysis

Population structure analysis and principal component analysis (PCA) were carried out using PLINK v1.9 [15] with the parameters (----indep--pair--wise wise 50 5 0.2). The genetic structure was estimated via admixture [16] with a kinship set from 2 to 4. Principal component analysis was conducted via smartPCA of the eigensoft v5.0 package [17]. A phylogenetic tree was constructed from 110 samples via PLINK via MEGA v7.0 [18] and visualized using itol (https://itol.embl.de/ accessed on 1 December 2024) [19].

VCFtools v0.1.17 [20] was used to estimate the nucleotide diversity (θπ) of each breed, keeping a window size of 50 kb and a step size of 20 kb. It was used to calculate the fixation index (*F*_ST_) between the six cattle breeds.

### 2.4. Detection of Selection Signals

In the present study, our aim was to identify regions exhibiting positive selection signatures in Zaosheng cattle. We detected the selection signatures within Zaosheng cattle via two different statistics: θπ and the composite likelihood ratio (CLR) [21]. Nucleotide diversity was estimated via a sliding window approach with windows of 50 kb and a step of 20 kb via VCFtools [20]. The CLR test was calculated for sites in nonoverlapping 50 kb windows by using SweepFinder 2 [22]. Empirical P values were calculated for the θπ and CLR windows, and the overlaps of the top 1% windows of each method were considered candidate signatures of selection.

Furthermore, the *F*_ST_, nucleotide diversity ratio (θπ ratio), and cross-population extended haplotype homozygosity (XPEHH) [23,24] were used to compare Zaosheng cattle (as a target population) with East Asian taurine cattle and East Asian indicine cattle (as a reference population). This cross-population analysis aimed to comprehensively identify shared selection signals among these distinct populations. *F*_ST_ and θπ ratio analyses were performed in 50 kb windows with 20 kb steps via VCFtools v0.1.16 [20]. XPEHH statistics were calculated for each population pair via Selscan v1.166 [25] We identified putative selective sweeps by selecting the top 1% of the original scores from each method, which represented the most likely candidates for the regions under selection. Tajima’s *D* statistic was computed via VCFtools for several important candidate genes.

### 2.5. Local Ancestry Inference

LOTER (https://github.com/bcm-uga/Loter accessed on 15 December 2024) [26] was used to infer taurine and indicine ancestry in the genomes of Zaosheng cattle. We selected the East Asian indicine and East Asian taurine groups as reference panels on the basis of population structure. The length and frequency of ancestral segments in each reference group were subsequently calculated. To detect a high proportion of fragments with ancestry, the ancestry-specific haplotypes for each fragment were compared to the total number of ancestry-specific haplotypes for all fragments, with regions of significance having a *p* value < 0.01 (Z test). The ideogram package 0.2.2 version [27] in R 4.3.2 version was used to draw chromosome maps to visualize excessive segments of East Asian indicine and East Asian taurine on the basis of the *B. taurus* reference genome. Functional enrichment analysis was performed on the list of genes within the excessive segments detected by KOBAS v3.0 [28].

## 3. Results

### 3.1. Data Collection, Sequencing, and Identification of SNPs

The DNA sequences of 110 genomes were used in this study, with an average sequence coverage of ~9.1 × and an average mapping rate of 99.69% (Appendix A). A total of 46,329,833 allelic autosomal SNPs were detected. The functional annotation of the polymorphic loci revealed that the majority of the SNPs were present in intergenic (38.0%) or intronic regions (58.9%). Exons represented 0.8% of the total SNPs, 93,688 of which were nonsynonymous SNPs and 142,001 of which were synonymous SNPs. Appendix A shows the total number of SNPs detected in each breed; East Asian indicine cattle had the highest number of SNPs, followed by crossbred Zaosheng cattle and Qinchuan cattle, and taurine cattle had the lowest number of SNPs (Appendix A).

### 3.2. Population Structure and Genetic Diversity

To explore the correlations between Zaosheng cattle and other cattle breeds distributed worldwide, we conducted admixture, neighbor-joining (NJ) tree and principal component analyses (PCAs) on autosomal genomic SNPs (Figure 1). The analyses revealed clear geographical patterns between the cattle populations. In the admixture analysis, the cattle breeds were divided into taurine or indicine ancestry at K = 2; whereas, at K = 3, East Asian taurine cattle were separated from European taurine cattle, while South Asian indicine cattle were further separated at K = 4. The PCA revealed a clear genetic structure, with samples from each geographical region clustering together. The first PC explained 6.19% of the total genetic variation and was driven by considerable genetic distance between *B. taurus* and *B. indicus*. The second PC explained 2.58% of the total variation and geographically separated the different indicine groups, such as East Asian indicine cattle and South Asian indicine cattle (Figure 1b). NJ tree analysis was used to construct a phylogenetic tree based on genetic distances, both of which corroborated the findings from the ADMIXTURE analysis (Figure 1c,e). Here, Qinchuan cattle and Zaosheng cattle clustered together. To explore the affinities between Zaosheng cattle and other animals, we constructed a smart PCA-based *F*_ST_ matrix, which depicted decreased *F*_ST_ values between Zaosheng cattle and Qinchuan cattle, revealing their close genetic identity (Figure 1d). The shortest distance was observed between Zaosheng cattle and Central Chinese cattle (Qinchuan cattle) (0.006); whereas, the farthest distance was observed between South Asian indicine cattle and Central Chinese cattle (0.205).

### 3.3. Genome-Wide Selective Scanning Signals from Zaosheng Cattle

#### 3.3.1. Genetic Signature of Selection in Zaosheng Cattle

θπ and CLR methods were applied to detect the genomic regions related to selection in the Zaosheng breed. The two methods yielded outlier signals (top 1%) in overlapping regions and were, therefore, considered candidate-selective regions. A total of 1388 (θπ) and 530 (CLR) genes with selection signatures in Zaosheng cattle were identified (Appendix A), 336 of which overlapped (Appendix A). Among these overlapping genes, *ACSS2* [29], NCOA6 [30], ASIP [31] were related to lipids, including lipid metabolism, fatty acid synthesis, and lipid deposition. *SLAMF1* [32] and *ROMO1* [33] play important roles in inflammation and the immune response, while GOLGA4 [34] and TP53INP2 [35] is important for fertility and reproductive traits. DCLK3 [36] was found to be involved in heat stress.

#### 3.3.2. Selective Signals Between Zaosheng Cattle and East Asian Indicine Cattle

Further elucidation of the positive selective sweep regions was acquired through *F*_ST_, the *θ*π ratio, and XPEHH between Zaosheng cattle and East Asian indicine cattle (Figure 2a–c). The genomic regions identified (*p* < 0.005) by at least two methods were considered candidate regions of positive selection (Appendix A). This multimethod approach increased the robustness of our identification of selection signatures. Candidate regions were annotated to 425 candidate genes (Appendix A). The functions of genes under positive selection included immunity (*HYAL1* and *HYAL2*) [37], phenotype (ASIP), fat metabolism and deposition (*TP53INP2*, *NCOA6*, and *ACSS2*), and quality of meat (*IGF1R* and *PLAGL2*) [38,39] and (*GSS*) [40], suggesting their strong selection in Zaosheng cattle.

We found that the regions of the *HYAL2* and *NOCA6* genes in Zaosheng cattle showed high F_ST_ values and low Tajima’s D value. The haplotype heatmaps of the *HYAL2* and *NOCA6* genes indicated significant differences between Zaosheng cattle and East Asian indicine cattle (Figure 2d,e). We subsequently performed functional enrichment analysis of the candidate genes via KEGG pathway analysis (Appendix A). The results revealed significant enrichment of two KEGG pathway terms (corrected *p* value < 0.01), including glycosaminoglycan degradation (corrected *p* value = 0.0000013) and metabolic pathways (corrected *p* value = 0.0026).

#### 3.3.3. Selection Signature Between Zaosheng Cattle and East Asian Taurine Cattle

We also implemented three methods to further elucidate the positive selection characteristics between Zaosheng cattle and East Asian taurine cattle (Figure 3a–c). The genomic regions identified (*p* < 0.005) by at least two methods were considered candidate regions of positive selection (Appendix A). A total of 153 candidate genes were identified, including genes related to immunity and growth (*SLAMF1*) [41]. Genes related to feed efficiency and meat quality traits (*PPP3R1*) were identified as candidate genes under selection [29,42]. Genes related to fertility (*PROKR1*) [43] and heat stress (*USH2A*) [44] adaptation were also found to be under selection pressure. Moreover, the high signal values of these genes revealed that Zaosheng cattle are potential local breeds and are subject to selection in a variety of areas, such as fat deposition, fertility, and immunity.

#### 3.3.4. Local Ancestry Inference of Zaosheng Cattle

Zaosheng cattle are considered to have a hybrid origin of indicine and taurine cattle. To infer the local taurine and indicine ancestries across the Zaosheng cattle genomes, LOTER was employed (Figure 4a, Appendix A) [26]. East Asian indicine cattle (*n* = 20) and East Asian taurine cattle (*n* = 20) were selected as reference samples. The segments with frequencies of at least 0.85 and lengths of at least 1000 bp were regarded as high-frequency ancestral fragments (*p* < 0.01). A total of 680 East Asian indicine segments and 5686 East Asian taurine segments were retained. The maximum lengths of the retained segments in the East Asian indicine and East Asian taurine groups were 112,449,384 bp and 112,447,365 bp, respectively.

The retained East Asian taurine segments in Zaosheng cattle contained 442 annotated genes. These genes were subjected to KEGG pathway enrichment analysis (corrected *p* value < 0.05). These genes were enriched in various pathways, including metabolic, phospholipase D signaling, cGMP−PKG signaling, vascular smooth muscle contraction, and retrograde endocannabinoid signaling pathways (Appendix A). By overlapping these annotated genes with the candidate genes of Zaosheng cattle (336 candidate genes were annotated through θπ and CLR methods), we identified 129 shared genes, including those related to immunity (*SLAMF1*, *CD48*, *CD84,* and *ROMO1*) [33,41,45,46], reproduction (*ADAMTSL1* and *LEO1*) [47,48], muscle production (*TMOD3*, *MYLK3,* and *TMOD2*) [49,50,51], and fat deposition (*ACSS2*, *ASIP*, *NCOA6,* and *TP53INP2*) [29,30,31,35]. Moreover, *NCSTN* and *GALNTL6* were found to potentially be associated with growth traits [52,53].

In our study, we found that the regions of the *TP53INP2* gene in Zaosheng cattle presented significantly higher *F*_ST_ values and lower Tajima’s *D* values than did the East Asian indicine cattle and exhibited a high degree of genetic overlap with taurine cattle. Moreover, the haplotypes of the *TP53INP2* genes in Zaosheng cattle originated from East Asian taurine ancestry (Figure 4c).

Moreover, the retained East Asian indicine segments in Zaosheng cattle contained 95 annotated genes. Functional enrichment analysis, specifically KEGG pathway enrichment analysis, was performed on these genes. The only significantly enriched KEGG pathway was “cell adhesion molecules (CAMs)” (corrected *p* value < 0.05) (Figure 4b, Appendix A). Zaosheng cattle have a hybrid origin involving indicine and taurine cattle, and haplotypes of indicine or taurine ancestry may confer a relative adaptive advantage under selection pressures. By overlapping these annotated genes with the candidate genes of Zaosheng cattle (336 candidate genes were annotated through θπ and CLR methods), a total of 14 genes were involved in growth (*GBP4*) [54], reproduction (*TUSC3*) [55], and heat stress adaptation (*USH2A*) [44]. We found that the PROKR1 gene was subject to selection in Zaosheng cattle and that a high proportion of indicine segment pedigrees were affected (Figure 4d). Moreover, through haplotype maps, we found that PROKR1 was selected in Zaosheng cattle.

## 4. Discussion

### 4.1. Genetic Ancestry and Population Structure of Zaosheng Cattle

Chinese indigenous cattle are characterized by their rich genetic resources, wide distribution range and diverse ecological types harboring favorable genetic traits [8]. Zaosheng cattle constitute a local breed in Gansu Province. The current study explored the population genetic structure of Zaosheng cattle. As indicated by the ADMIXTURE analysis, the ancestral contributions of Zaosheng cattle were East Asian indicine (~25.9%), East Asian taurine (~72.5%), and European taurine (~0.16%). In ancient times, the Zaosheng area was remote, and to develop agriculture, Qinchuan cattle were introduced from the Guanzhong area of Shaanxi, where agriculture was well-developed, and these cattle were bred. In recent years, with improvements in quality of life, the requirements for the quantity and quality of beef have increased [8]. Qinchuan cattle are a representative breed of Chinese native cattle, but since the 1990s, the number of Qinchuan cattle has decreased sharply. We conducted an in-depth study of the genomic information of Zaosheng cattle and reported that Zaosheng cattle are genetically closely related to Qinchuan cattle by *F*_ST_. We found that cattle breeds with equivalent beef qualities could be used to help develop the market.

### 4.2. Selection Signatures and Environmental Adaptation

The selection signal analysis of Zaosheng cattle in this study provides new insights into the genetic basis of adaptations to the local environment. Here, the θπ and CLR methods were employed to detect selected genes in Zaosheng cattle, and some genes were strongly selected. For example, ROS modulator 1 (*ROMO1*) is a mitochondrial membrane protein that is essential for the regulation of mitochondrial ROS production and redox sensing and has been shown to play an important role in the regulation of mitochondrial ROS production and redox sensing [56]. Oxidative stress can be induced by various environmental factors, such as heat stress, disease, or injury [56]. Therefore, *ROMO1* may help in modulating the response to such stresses, protecting cells from damage and promoting recovery. Recent studies have shown that the overexpression of Romo1 increases anti-inflammatory function and promotes the reprogramming of the cellular metabolism of macrophages [57]. While *ROMO1* plays a crucial role in mitochondrial function and stress response, another gene of interest, *GOLGA4*, is involved in cellular structural organization and reproductive biology. *GOLGA4* is a Golgi matrix protein, and recent studies have shown that *GOLGA4* is also expressed in mouse testes [58,59].

Given that Zaosheng cattle are primarily of East Asian indicine and East Asian taurine ancestry, we aimed to determine the role of East Asian indicine and East Asian taurine ancestors on Zaosheng cattle characteristics. Our analyses revealed the answer: the East Asian indicine lineage was found to predominantly influence growth-related traits (mediated by *PROKR1*), while the East Asian taurine lineage primarily contributed to meat quality characteristics (associated with *TP53INP2*). A comparative analysis was subsequently conducted between Zaosheng cattle and East Asian indicine cattle to identify advantageous genes specific to Zaosheng cattle relative to East Asian indicine cattle. The analysis revealed several genes involved in the immune system, particularly the *HYAL1* and *HYAL2* genes, which overlapped among the three selection methods. Tajima’s *D* also revealed that *HYAL2* was indeed selected between Zaosheng cattle and East Asian indicine cattle. In general, East Asian indicine cattle, exemplified by Hainan cattle, have good immune performance [60], but we identified clusters of immune genes in Zaosheng cattle that presented different haplotypes (Figure 2d). These findings indicate that Zaosheng cattle and East Asian indicine cattle may have different immune mechanisms, presumably due to environmental differences. Hyaluronidase (*HYAL*)-2 is a weak, acid-active, hyaluronan-degrading enzyme broadly expressed in somatic tissues [61]. Some studies have shown that deregulated hyaluronan metabolism in the tumor microenvironment drives cancer inflammation, but *HYAL2* has catabolic functions that can reduce excessive inflammation and contribute to the resolution of the immune response [62]. Several studies have confirmed that *NCOA6* deletion can lead to early embryonic death or slow growth in mice, potentially by disrupting the cell cycle and increasing apoptosis [63]. Thus, *NCOA6* may be involved in regulating bovine growth and development. In our study, we found that the haplotypes were different in Zaosheng cattle and East Asian indicine cattle in this region (Figure 2e), indicating the presence of selection pressure in this region.

Zaosheng cattle, a native Chinese breed, demonstrate remarkable resilience and environmental adaptability [9]. We detected higher signal values for genes related to immunity in Zaosheng cattle than in East Asian taurine cattle, indicating outstanding immunity in Zaosheng cattle. Previously, some researchers reported that the USH2A gene affects cattle hair color and suggested that it may be associated with adaptive traits or survival advantages [64]. *LIMCH1*, an actin stress fiber-associated protein, is a paralogous protein with C-terminal LIM domains [65]. A study examining myofibers affected by neurogenic muscular atrophy revealed the overexpression of 55 proteins found that most of them, including *LIMCH1* were involved in myofibrillogenesis [61]. In cattle, a study concluded that the *LIMCH1* locus is a putative region underlying greater forehead size in Brahman cattle than in Yunling cattle [66]. Taken together, *LIMCH1* may affect skeletal development and, subsequently, body size in cattle.

### 4.3. Local Ancestry Inference of Zaosheng Cattle

In our study, we applied LOTER and selection analysis to infer local ancestry and obtain ancestral selection signatures in Zaosheng cattle. These excessive East Asian taurine segment-annotated genes were enriched mainly in growth, metabolism, development, and disease pathways. For a few East Asian indicine segments, the annotated genes were enriched mainly in cell adhesion molecules. Cell adhesion molecules, including receptors of the immunoglobulin superfamily and integrins, particularly integrins, play a vital role in regulating all aspects of immune cell function [67].

Genes related to fat metabolism and muscle production were more common in East Asian taurine cattle and Zaosheng cattle. The high muscle fat content and better taste of beef from East Asian taurine cattle, such as Yanbian cattle [68], indicate that the target group of Zaosheng cattle has excellent meat quality. Owing to the limited number of East Asian indicine descent segments and fewer annotated genes in these segments, the overlap with Zaosheng cattle candidate genes was substantially reduced. Interestingly, the *USH2A* gene is also a candidate gene for interspecific selection between Zaosheng and East Asian taurine cattle; we believe that Zaosheng cattle also have strong environmental adaptability. Notably, the *PROKR1* gene emerged as a common candidate across all three analytical approaches, as follows: the annotated genes from East Asian indicine descent segments, the annotated genes from the comparison of Zaosheng cattle and East Asian taurine cattle, and the candidate genes from Zaosheng cattle. *PROK1* is also termed endocrine gland-derived vascular endothelial growth factor (endocrine gland-derived VEGF). Studies in pigs have shown that *PROK1*, which acts via *PROKR1*, promotes the formation of capillary-like structures via endothelial cells isolated from porcine corpus luteum in vitro and stimulates the synthesis of VEGFA and the mRNA expression of angiogenin in the corpus luteum [69]. Prokineticin 1, the protein encoded by the *PROK1* gene, is a novel factor that regulates porcine corpus luteum function [43]. We further analyzed this genomic region and found that this gene was subjected to strong selection in Zaosheng cattle. We hypothesized that it originated from the East Asian indicine population, but after long-term evolution, it also presented unique characteristics.

In our study, *TP53INP2* emerged as a significant candidate gene under selection in Zaosheng cattle, showing a haplotype pattern identical to East Asian taurine cattle but distinct from East Asian indicine cattle, suggesting taurine lineage influence. This finding aligns with its potential role in meat quality traits. The tumor protein p53-induced nuclear protein 2 (*TP53INP2*) is a multifunctional regulator of apoptosis, autophagy, and cell differentiation [70]. In skeletal muscle specifically, TP53INP2 modulates muscle mass in adult mice through autophagy regulation, where its overexpression leads to increased autophagy and moderately reduced fiber size [71]. Notably, human studies correlate elevated TP53INP2 levels with improved muscle strength, physical performance, and healthy aging [71]. These studies strongly support the contribution of *TP53INP2* to the quality meat characteristics observed in Zaosheng cattle.

The unique genetic makeup of Zaosheng cattle (25.9% East Asian indicine, 72.5% East Asian taurine, and 0.16% European taurine) confers distinct biological advantages, including adaptation to local environments. Chinese ancestry contributes to heat tolerance via the *ROMO1*-mediated oxidative stress response and disease resistance through the *HYAL2*-associated immune mechanism. The dominant East Asian taurine lineage drives intramuscular fat deposition (PROKR1-regulated angiogenesis) and skeletal muscle development (*TP53INP2*-controlled autophagy). The minimal amount of European taurine introgression (0.16%) may reflect historical breeding practices. Thus, this hybrid ancestry exemplifies how balanced genetic composition can enhance both ecological adaptation and economic traits in indigenous cattle.

### 4.4. Conservation and Breeding Implications

The Zaosheng cattle breed is a valuable and widely used genetic resource in local areas because of its excellent meat quality and high adaptability. Given the prevalence of crossbreeding and the increasing depletion of indigenous cattle genetic resources, it is critical to design breeding programs that will improve and conserve China’s indigenous cattle. In this context, our results provide a basis for further research on the genomic characteristics of Zaosheng cattle in relation to economically important traits.

## 5. Conclusions

In conclusion, this study comprehensively investigated the genomic variations in Zaosheng cattle by screening whole-genome resequencing data. We explored the population structure of current Zaosheng cattle, elucidated their genetic diversity, conducted selective sweep analysis, and explored the contributions of different ancestral lineages to their genomes. Simultaneously, a set of potential candidate genes with potential impacts on fat deposition and development, meat quality, and the immune response were identified within this breed. These discoveries not only advance our knowledge of the unique characteristics of Zaosheng cattle but also provide a basis for genetic breeding and resource protection in Zaosheng cattle.

## Figures and Tables

**Figure 1 biology-14-00623-f001:**
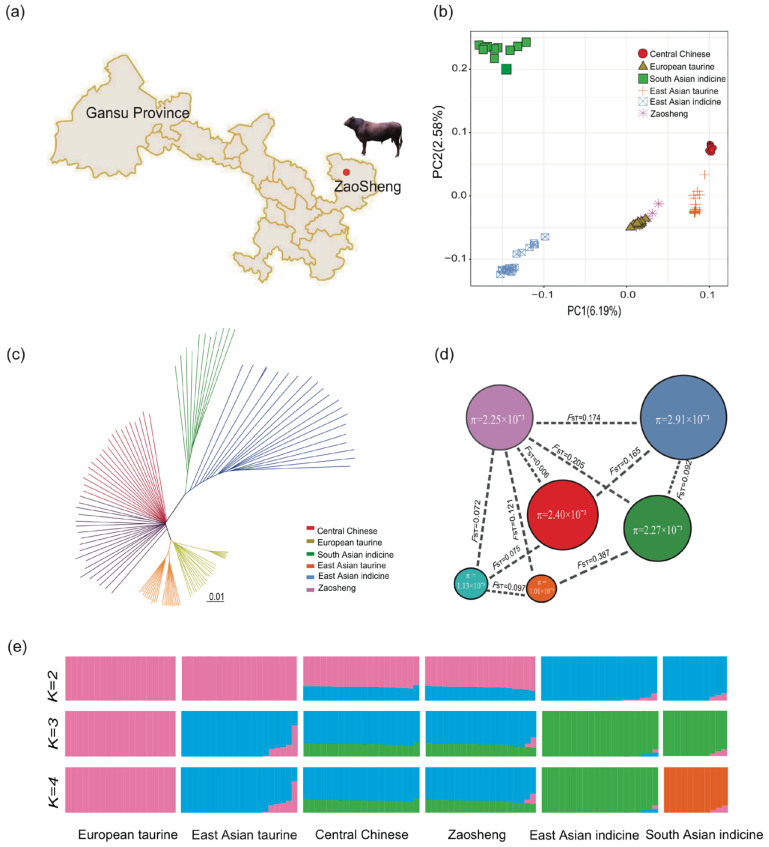
Genetic diversity of Zaosheng cattle. The color scales in panels (**b**–**d**) are consistent. (**a**) Distribution map of Zaosheng cattle; (**b**) principal component analysis of cattle populations with PC1 against PC2; (**c**) neighbor-joining tree of the relationships between Zaosheng cattle and possible ancestors; (**d**) genetic distances estimated between each population via the *F*_ST_; (**e**) model-based clustering of cattle breeds via ADMIXTURE. The breeds are colored according to their geographic region and labeled with breed names.

**Figure 2 biology-14-00623-f002:**
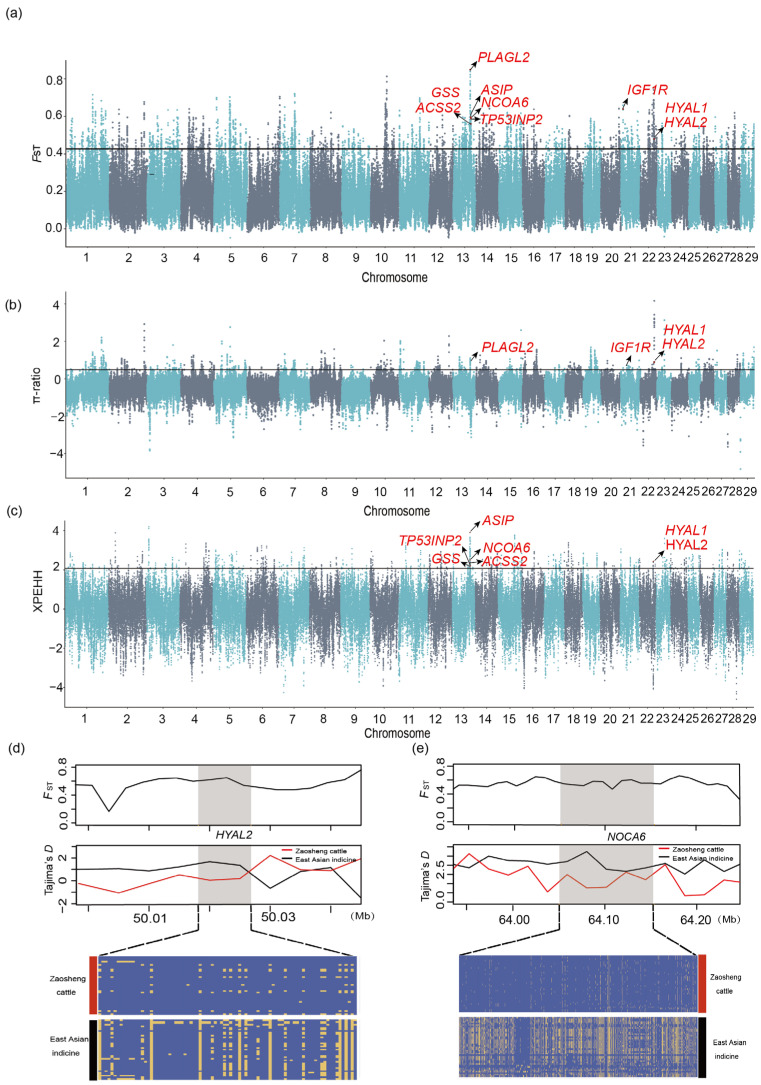
Genomic region with strong selective sweep signals in Zaosheng cattle and East Asian indicine. Manhattan plots against selective sweep analysis between Zaosheng cattle and East Asian indicine cattle with (**a**) *F*_ST_; (**b**) π-ratio; (**c**) XPEHH; (**d**) pairwise *F*_ST_ values, Tajima’s D value and haplotype pattern heatmap of the *HYAL2* gene region; (**e**) pairwise *F*_ST_ values, Tajima’s D value and haplotype pattern heatmap of the *NOCA6* gene region.

**Figure 3 biology-14-00623-f003:**
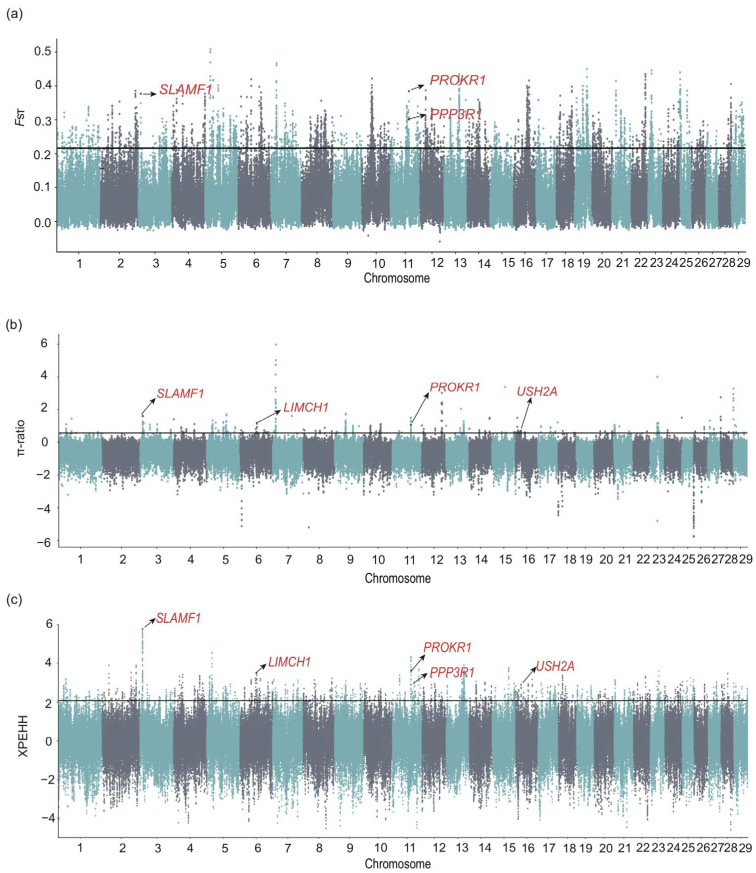
The genomic region with strong selective sweep signals in Zaosheng cattle and East Asian taurine cattle. Manhattan plots against selective sweep analysis between Zaosheng cattle and East Asian taurine cattle using (**a**) *F*_ST_, (**b**) the π ratio, and (**c**) XPEHH.

**Figure 4 biology-14-00623-f004:**
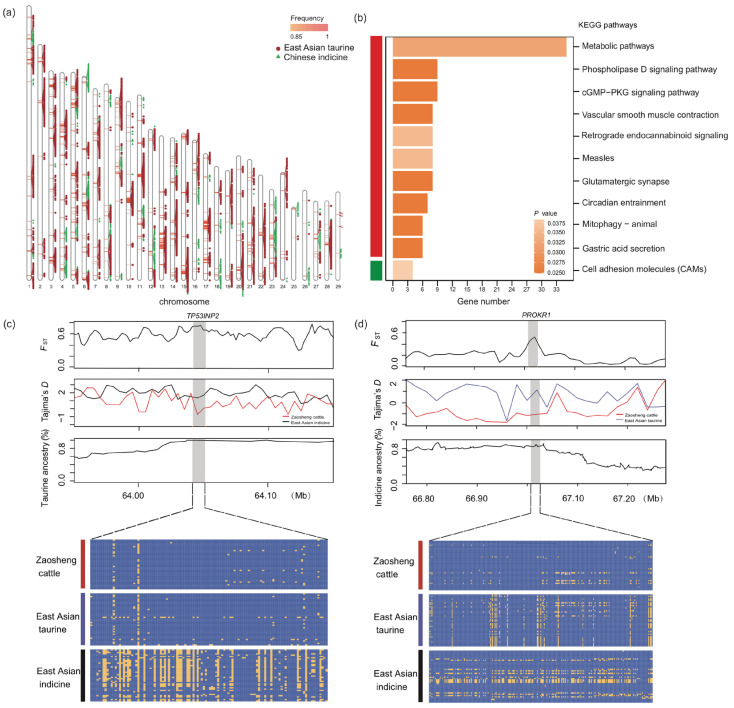
Identification of the local segments in which proportions of a certain ancestry were significantly greater than the proportion in the whole genome in Zaosheng cattle. (**a**) Distribution of the local segments with proportions of East Asian indicine and East Asian taurine ancestries. (**b**) KEGG pathway enrichment analysis of genes associated with excessive East Asian indicine proportions and East Asian taurine proportions. (**c**) Pairwise F_ST_, Tajima’s *D* value, haplotype pattern heatmap, and average taurine ancestry (%) of the *TP53INP2* gene region. (**d**) Pairwise *F*_ST_, Tajima’s *D* value, haplotype pattern heatmap, and average taurine ancestry (%) of the *PROKR1* gene region.

## Data Availability

The original data presented in the study are openly available in GenBank (PRJNA1085861). A preliminary version of this work was published in Research Square under the DOI: https://doi.org/10.21203/rs.3.rs-6059907/v1. The current manuscript includes additional data analyses, expanded results, and peer-reviewed revisions.

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
