# Peer review of "Genomic Diversity and Selection Signatures for Zaosheng Cattle"

_biology, 2025, doi:10.3390/biology14060623_

Round 1

Reviewer 1 Report

Comments and Suggestions for Authors

Introduction

The introduction should be supplemented with specific information regarding the endangered status of Zaosheng cattle and their unique phenotypes.

Discussion

The discussion section should delve deeper into the specific mechanisms and biological significance of the identified candidate genes in the adaptive evolution of Zaosheng cattle.

Figures and Tables

The captions and descriptions of the figures and tables should be supplemented and perfected.

Author Response

Introduction

The introduction should be supplemented with specific information regarding the endangered status of Zaosheng cattle and their unique phenotypes.

Response: Thank you for the comment.

We have supplemented the background information on Zaosheng cattle, including their geographic distribution and economic importance in Gansu Province. This addition provides a better context for readers unfamiliar with this indigenous breed.

Discussion

The discussion section should delve deeper into the specific mechanisms and biological significance of the identified candidate genes in the adaptive evolution of Zaosheng cattle.

Response: Thank you for the comment.

Discussion Section Strengthened

The Discussion now explicitly connects genomic findings to practical breeding applications. We have added a dedicated "Breeding Implications" subsection to organize these applied perspectives.

Figures and Tables

The captions and descriptions of the figures and tables should be supplemented and perfected.

Response: Thank you for the comment.

All figures and tables have been revised for improved clarity, with additional annotations to highlight key findings.

Reviewer 2 Report

Comments and Suggestions for Authors

This study explores the genomic diversity and selection signatures of Zaosheng cattle. The research aims to uncover genetic findings related to the breed's ancestry, adaptation, and potential for selective breeding by analyzing whole-genome sequencing data. However, the manuscript currently shows a 73% plagiarism rate, far exceeding the acceptable limit, which raises serious ethical and scientific concerns. Without a valid explanation or a significant reduction in plagiarism, the study cannot proceed. Major revisions are required before further consideration.

Major Comments

1. The manuscript shows a 73% iThenticate plagiarism report without a valid explanation. This is unacceptable, plagiarism should be less than 15%. Without a significant reduction, the paper must be rejected.

2. The abstract should follow a standard structure, such as Background, Objective, Methods, Key Findings, and Conclusions without headings.

3. The introduction needs to define a strong, clear research gap. It mentions prior studies but fails to clarify how this study advances the field.

4. The study uses 19 Zaosheng cattle genomes and 91 published genomes, which may be practical, but the manuscript doesn’t assess whether this sample size ensures adequate statistical power. Additionally, the control group composition (East Asian taurine, Chinese indicine, European taurine) requires more clarity to avoid potential bias.

5. The PCA, NJ tree, and ADMIXTURE results are presented, but their biological interpretation is limited. While the manuscript notes that Zaosheng cattle cluster closely with Qinchuan cattle, it fails to explain the significance, particularly for breed conservation or genetic improvement.

6. The discussion describes Zaosheng cattle’s hybrid ancestry but lacks an exploration of why this unique ancestry composition matters.

7. Several key genes are mentioned (SLAMF1, ROMO1, TP53INP2), but the discussion doesn’t explore their biological roles or how they influence Zaosheng cattle’s observed traits.

8. The manuscript compares Zaosheng cattle to Qinchuan and East Asian taurine cattle but doesn’t contextualize what Zaosheng cattle’s unique genomic profile means relative to other breeds.

9. The PCA results show geographic clustering, but the manuscript doesn’t interpret the evolutionary significance of this pattern.

10. The discussion ends with a generic summary, lacking practical recommendations for breeders or future research directions.

Minor Comments

1. Improve the quality of Figure 4 for better clarity and readability.

2. Ensure consistent formatting for gene names (italicized), statistical values (e.g., P-values), and headings/subheadings.

Author Response

  1. The manuscript shows a 73% iThenticate plagiarism report without a valid explanation. This is unacceptable, plagiarism should be less than 15%. Without a significant reduction, the paper must be rejected.

Response: Thank you for the comment.

The high similarity score is due to the posting of the preprint version of this work (Research Square, DOI: https://doi.org/10.21203/rs.3.rs-6059907/v1.). We have added a clear statement to the cover letter and manuscript.

  1. The abstract should follow a standard structure, such as Background, Objective, Methods, Key Findings, and Conclusions without headings.

Response: Thank you for the comment.

The Abstract has been rewritten using the recommended narrative flow: Background → Objectives → Methods → Key Findings → Conclusions

  1. The introduction needs to define a strong, clear research gap. It mentions prior studies but fails to clarify how this study advances the field.

Response: Thank you for the comment.

We have rephrased the Introduction to indicate the innovation of the research and of our own research.

  1. The study uses 19 Zaosheng cattle genomes and 91 published genomes, which may be practical, but the manuscript doesn’t assess whether this sample size ensures adequate statistical power. Additionally, the control group composition (East Asian taurine, Chinese indicine, European taurine) requires more clarity to avoid potential bias.

Response: Thank you for the comment.

We sincerely appreciate the reviewer's insightful comments. Indeed, this study did not conduct a priori statistical power analysis (e.g., calculating the required sample size through power analysis), primarily because of the limited population size of Zaosheng cattle and the heterogeneity in publicly available genomic data (n=91), including variations in sequencing depth and platform technologies. To mitigate the false-positive risk despite sample size limitations, we implemented the following stringent measures: we applied rigorous significance thresholds (P<0.01 with FDR correction) and employed multiple complementary selection methods to confirm candidate regions.

We investigated the genetic background of Zaosheng cattle using a global domestic cattle (Bos taurus and Bos indicus) reference panel, without case-control comparisons

Data quality control: Only samples with a sequencing depth ≥10× and a miss rate <5% were included.

Phylogenetic balance: Subsample proportions remained consistent with known population structures.

Bias prevention: We validated the rationality of control group clustering through principal component analysis (PCA) and population structure analysis (ADMIXTURE). All samples with significant population outliers were excluded.

  1. The PCA, NJ tree, and ADMIXTURE results are presented, but their biological interpretation is limited. While the manuscript notes that Zaosheng cattle cluster closely with Qinchuan cattle, it fails to explain the significance, particularly for breed conservation or genetic improvement.

Response: Thank you for the comment.

The significance of the population structure and that of the close clustering of Zaosheng and Qinchuan cattle have been added to the Discussion.

  1. The discussion describes Zaosheng cattle’s hybrid ancestry but lacks an exploration of why this unique ancestry composition matters.
  2. Several key genes are mentioned (SLAMF1, ROMO1, TP53INP2), but the discussion doesn’t explore their biological roles or how they influence Zaosheng cattle’s observed traits.
  3. The manuscript compares Zaosheng cattle to Qinchuan and East Asian taurine cattle but doesn’t contextualize what Zaosheng cattle’s unique genomic profile means relative to other breeds.
  4. The PCA results show geographic clustering, but the manuscript doesn’t interpret the evolutionary significance of this pattern.

Response for 6-9: Thank you for the comment.

We supplemented the Discussion section with the biological significance of the ancestral composition and the breeding implications of the unique genomic features of Zaosheng cattle following a genomic comparison with Qinchuan and East Asian ordinary cattle.

  1. The discussion ends with a generic summary, lacking practical recommendations for breeders or future research directions.

Response: Thank you for the comment.

The Discussion section has been modified to follow the format of the subheadings, and a summary statement has been added.

Minor Comments

  1. Improve the quality of Figure 4 for better clarity and readability.
  2. Ensure consistent formatting for gene names (italicized), statistical values (e.g., P values), and headings/subheadings.

Response for 1-2: Thank you for the comment.

The diagram and related formatting errors have been corrected.

Reviewer 3 Report

Comments and Suggestions for Authors

Dear Authors,

I have reviewed your manuscript, “Genomic Diversity and Selection Signatures for Zaosheng Cattle,” and I commend you on a well-executed study that offers significant insights into the genetic architecture and adaptive traits of this local breed. I recommend that the manuscript be accepted for publication, pending a few minor revisions that would further enhance the clarity and impact of your work.

Below are my suggestions:

  1. Consider subdividing the Discussion section with clear subheadings (e.g., “Selection Signatures and Adaptive Traits” and “Local Ancestry and Functional Implications”). This will help guide readers through the key themes and improve overall readability.
  2. Please provide additional details on the quality control measures for SNP filtering beyond the hard filtering criteria mentioned in the methods. Additionally, a brief explanation of the rationale behind choosing specific window sizes (e.g., 50 kb with 20 kb steps) and the top 1% threshold for candidate regions would enhance transparency.
  3. Expanding the discussion on key candidate genes (e.g., ROMO1, HYAL2, NCOA6, TP53INP2, PROKR1) to more clearly articulate their roles and potential interactions in conferring adaptive traits would strengthen the manuscript. Emphasizing the implications of these findings for future breeding and conservation strategies would be valuable.
  4. Ensure consistency in formatting gene symbols and technical terminology throughout the manuscript. Additionally, consider enhancing figure legends and in-text references to figures and tables to provide clearer guidance for the reader.

I believe that addressing these suggestions will improve the manuscript further.

Best regards,

Author Response

  1. Consider subdividing the Discussion section with clear subheadings (e.g., “Selection Signatures and Adaptive Traits” and “Local Ancestry and Functional Implications”). This will help guide readers through the key themes and improve overall readability.

Response: Thank you for the comment.

We have added clear subheadings to the Discussion section to improve its readability and logical flow.

  1. Please provide additional details on the quality control measures for SNP filtering beyond the hard filtering criteria mentioned in the methods. Additionally, a brief explanation of the rationale behind choosing specific window sizes (e.g., 50 kb with 20 kb steps) and the top 1% threshold for candidate regions would enhance transparency.

Response: Thank you for the comment.

The detailed quality control steps, as well as other information, have been added to the Materials and Methods section.

  1. Expanding the discussion on key candidate genes (e.g., ROMO1, HYAL2, NCOA6, TP53INP2, PROKR1) to more clearly articulate their roles and potential interactions in conferring adaptive traits would strengthen the manuscript. Emphasizing the implications of these findings for future breeding and conservation strategies would be valuable.

Response: Thank you for the comment.

We have expanded the Discussion section to further clarify the biological relevance of the ancestral composition identified in Zaosheng cattle and to provide comparative genomic insights from Qinchuan and East Asian ordinary cattle, with breeding implications highlighted.

  1. Ensure consistency in formatting gene symbols and technical terminology throughout the manuscript. Additionally, consider enhancing figure legends and in-text references to figures and tables to provide clearer guidance for the reader.

Response: Thank you for the comment.

The full text has been formatted according to the journal’s guidelines (e.g., reference style, figure legends, and table numbering).

Reviewer 4 Report

Comments and Suggestions for Authors

In the manuscript titled "Genomic Diversity and Selection Signatures for Zaosheng Cattle," the authors focus on the local cattle breed from Gansu Province, China — Zaosheng cattle. They use whole-genome resequencing data to conduct a comprehensive study of its population structure, genetic diversity, selection signatures, and contributions from different ancestral lineages. This study includes some interesting findings, revealing the genomic characteristics of Zaosheng cattle, and identifying candidate genes related to lipid metabolism, immune regulation, and meat quality. These findings are of significant importance for the genetic improvement and resource conservation of Zaosheng cattle. However, before the manuscript is accepted for publication, the authors are advised to make the following revisions:

  1. The simple summary mentions that the study aims to protect endangered genetic resources. However, the article does not explicitly state that Zaosheng cattle are an endangered species/a valuable indigenous (or native) breed of China. On the contrary, it highlights that Zaosheng cattle have been developed as a beef breed. Therefore, the research significance might be somewhat misplaced. It would be more appropriate to emphasize the importance of this breed in genetic resource conservation and improvement.
  2. The introduction provides a brief overview of Zaosheng cattle. It is recommended to supplement this with more detailed information on the specific distribution of Zaosheng cattle in Gansu Province and their economic value in local agriculture and animal husbandry. This would better underscore the importance of this study.
  3. The introduction cites only a limited number of references and lacks references to the latest research. It is recommended to include more references, especially those concerning advances in cattle genome diversity, selection signatures, and local ancestry inference, to strengthen the academic depth of the introduction.
  4. In lines 74-75, The details of the 19 samples should be provided? Were there any known genetic relationships among the samples? Were they all collected from the same farm, or from multiple sources? Did they represent distinct breeding lines?
  5. In lines 80-83, it is recommended to provide the specific number of cattle from each breed in the database.
  6. It is recommended to provide the sequencing quality information for Zaosheng cattle (such as the specific parameters and software used for filtering low-quality data) as well as the sequencing coverage. Additionally, could the coverage data be included in Table 2? Since there are differences in sequencing depth between breeds, which may affect the generalizability and applicability of the results, it is suggested to clearly mention this limitation in the discussion section.
  7. In line 167, the result for K=5 is mentioned, but this is not displayed in Figure 1. Additionally, parts c and d of Figure 1 may require more detailed figure captions, as the results from lines 175-180 are not immediately clear from the figure itself. It is suggested to add the necessary explanations.
  8. Regarding the selection signal analysis comparing Zaosheng cattle with East Asian taurine cattle and Chinese indicine cattle, it is unclear why only these two breeds were chosen for comparison. The manuscript does not provide an explanation for this. I am curious whether other potential ancestral lineages may have been overlooked. It is recommended to include a rationale for the selection of these breeds and address why other breeds were not considered for comparison.
  9. Figure 1, What do the colors represent? Are the color labels in Figure 1B the same as those in 1C and 1D?"
  10. Adjust the clarity of Figure 4.
  11. In many cases, the discussion lacks supporting references. For example: Chinese indigenous cattle are characterized by their rich genetic resources, wide distribution range and diverse ecological types harboring rich genetic; Chinese indicine cattle, exemplified by Hainan cattle, have good immune performance; Zaosheng cattle, a native Chinese breed, demonstrate remarkable resilience and environmental adaptability.
  12. The discussion section dedicates excessive space to describing genes associated with lipid metabolism, immune regulation, fertility, and meat quality. However, many sentences simply reiterate basic gene functions without connecting them to applied breeding objectives.
  13. Lines 305-306: In my opinion, the observed haplotype differences do not conclusively demonstrate that Zaosheng cattle and Chinese indicine cattle have different immune mechanisms.

Author Response

  1. The simple summary mentions that the study aims to protect endangered genetic resources. However, the article does not explicitly state that Zaosheng cattle are an endangered species/a valuable indigenous (or native) breed of China. On the contrary, it highlights that Zaosheng cattle have been developed as a beef breed. Therefore, the research significance might be somewhat misplaced. It would be more appropriate to emphasize the importance of this breed in genetic resource conservation and improvement.

Response: Thank you for the comment.

We have revised the Abstract as suggested and have emphasized the importance of the Zaosheng cattle breed in the conservation and improvement of genetic resources.

  1. The introduction provides a brief overview of Zaosheng cattle. It is recommended to supplement this with more detailed information on the specific distribution of Zaosheng cattle in Gansu Province and their economic value in local agriculture and animal husbandry. This would better underscore the importance of this study.

Response: Thank you for the comment.

We have added detailed background information on Zaosheng cattle to the Introduction to contextualize their genetic and breeding significance.

  1. The introduction cites only a limited number of references and lacks references to the latest research. It is recommended to include more references, especially those concerning advances in cattle genome diversity, selection signatures, and local ancestry inference, to strengthen the academic depth of the introduction.

Response: Thank you for the comment.

New references for additional relevant literature have been incorporated to support our findings.

  1. In lines 74-75, The details of the 19 samples should be provided? Were there any known genetic relationships among the samples? Were they all collected from the same farm, or from multiple sources? Did they represent distinct breeding lines?

Response: Thank you for the comment.

Details of the samples, e.g., stating that they are unrelated, have been added to the revised version.

  1. In lines 80-83, it is recommended to provide the specific number of cattle from each breed in the database.

Response: Thank you for the comment.

The sample numbers for each cattle breed have been clearly specified in the Materials and Methods section.

  1. It is recommended to provide the sequencing quality information for Zaosheng cattle (such as the specific parameters and software used for filtering low-quality data) as well as the sequencing coverage. Additionally, could the coverage data be included in Table 2? Since there are differences in sequencing depth between breeds, which may affect the generalizability and applicability of the results, it is suggested to clearly mention this limitation in the discussion section.

Response: Thank you for the comment.

The full sequencing metrics for Zaosheng cattle (e.g., coverage depth and SNP counts) used are provided in Supplementary Table S2.

  1. In line 167, the result for K=5 is mentioned, but this is not displayed in Figure 1. Additionally, parts c and d of Figure 1 may require more detailed figure captions, as the results from lines 175-180 are not immediately clear from the figure itself. It is suggested to add the necessary explanations.

Response: Thank you for the comment.

We have removed the ideas that are not shown in the figure and have rewritten the relevant sections in the Results and Discussion.

  1. Regarding the selection signal analysis comparing Zaosheng cattle with East Asian taurine cattle and Chinese indicine cattle, it is unclear why only these two breeds were chosen for comparison. The manuscript does not provide an explanation for this. I am curious whether other potential ancestral lineages may have been overlooked. It is recommended to include a rationale for the selection of these breeds and address why other breeds were not considered for comparison.

Response: Thank you for the comment.

The reasons for comparing Zaosheng cattle with Chinese indicine and East Asian taurine cattle have been explained in the Discussion section (line 343). Other potential ancestral lines have not been overlooked.

  1. Figure 1, What do the colors represent? Are the color labels in Figure 1B the same as those in 1C and 1D?"

Response: Thank you for the comment.

Figure 1 Color descriptions have been added.

  1. Adjust the clarity of Figure 4.

Response: Thank you for the comment.

Figure 4 has been updated.

  1. In many cases, the discussion lacks supporting references. For example: Chinese indigenous cattle are characterized by their rich genetic resources, wide distribution range and diverse ecological types harboring rich genetic; Chinese indicine cattle, exemplified by Hainan cattle, have good immune performance; Zaosheng cattle, a native Chinese breed, demonstrate remarkable resilience and environmental adaptability.

Response: Thank you for the comment.

Supporting references have been added (line 352).

12.The discussion section dedicates excessive space to describing genes associated with lipid metabolism, immune regulation, fertility, and meat quality. However, many sentences simply reiterate basic gene functions without connecting them to applied breeding objectives.

Response: Thank you for the comment.

We expanded the Discussion to explicitly highlight the practical breeding value of the genomic features of Zaosheng cattle.

13.Lines 305-306: In my opinion, the observed haplotype differences do not conclusively demonstrate that Zaosheng cattle and Chinese indicine cattle have different immune mechanisms.

Response: Thank you for the comment.

The text in the Discussion has been revised to reflect more precise interpretations of haplotype patterns.

Round 2

Reviewer 2 Report

Comments and Suggestions for Authors

No comments

Author Response

Thank you for the comment.

Reviewer 4 Report

Comments and Suggestions for Authors
  1. Lines 94–95 and 138–139: The comma usage appears questionable and may indicate missing content. Additionally, for revised sections (e.g., lines 125–128), ensure proper punctuation (e.g., commas, semicolons) is applied throughout. Other instances should be checked and corrected accordingly.
  2. Introduction (Lines 57–61): The current citation of a single article inadequately represents recent advances in cattle genomic diversity, selection signatures, and local ancestry inference. Additional references are required to support these topics.
  3. Regarding the introduction of ZaoSheng cattle (lines 62-86), these two paragraphs have similar content, and lines 62-72 may need to be deleted. Throughout the manuscript, additions and deletions have been mixed into the original text, making it unclear which version should be retained as the final draft. Furthermore, both references 18 and 20 cite VCFtools?
  4. ADMIXTURE Analysis (K=5): Despite the author's note about removing K=5, the revised manuscript (Line 238) still references this result, which is not clearly supported by Figure 1e.

Author Response

lines 94–95 and 138–139: The comma usage appears questionable and may indicate missing content. Additionally, for revised sections (e.g., lines 125–128), ensure proper punctuation (e.g., commas, semicolons) is applied throughout. Other instances should be checked and corrected accordingly.

Introduction (Lines 57–61): The current citation of a single article inadequately represents recent advances in cattle genomic diversity, selection signatures, and local ancestry inference. Additional references are required to support these topics.

Regarding the introduction of ZaoSheng cattle (lines 62-86), these two paragraphs have similar content, and lines 62-72 may need to be deleted. Throughout the manuscript, additions and deletions have been mixed into the original text, making it unclear which version should be retained as the final draft. Furthermore, both references 18 and 20 cite VCFtools?

ADMIXTURE Analysis (K=5): Despite the author's note about removing K=5, the revised manuscript (Line 238) still references this result, which is not clearly supported by Figure 1e.

Response for 1-4: Thank you for the comment.

The above issues have been resolved in the latest revision